# Optimizing ALT Flap Harvest: The Role of Combined Preoperative Duplex Ultrasound and Intraoperative ICG Angiography for Perforator Selection

**DOI:** 10.3390/life15040620

**Published:** 2025-04-07

**Authors:** Benjamin Thomas, Florian Falkner, Oliver Didzun, Adriana C. Panayi, Iman Ghanad, Gabriel Hundeshagen, Emre Gazyakan, Ulrich Kneser, Amir K. Bigdeli

**Affiliations:** 1BG Klinik Ludwigshafen, Department of Hand, Plastic, and Reconstructive Surgery, Burn Centre at Heidelberg University, Ludwig-Guttmann-Str. 13, 67071 Ludwigshafen, Germany; benjamin.thomas@bgu-ludwigshafen.de (B.T.); florian.falkner@bgu-ludwigshafen.de (F.F.); oliver.didzun@googlemail.com (O.D.); adriana.panayi@icloud.com (A.C.P.); iman.ghanad@gmail.com (I.G.); gabriel.hundeshagen@bgu-ludwigshafen.de (G.H.); emre.gazyakan@bgu-ludwigshafen.de (E.G.); ulrich.kneser@bgu-ludwigshafen.de (U.K.); 2Department of Plastic, Reconstructive, Aesthetic and Hand Surgery, Klinikum Kassel, Teaching Hospital of Philipps University Marburg, Mönchebergstraße 41-43, 34125 Kassel, Germany

**Keywords:** ALT, ICG, duplex, perforator flap, donor-site morbidity

## Abstract

Planning and harvesting anterolateral thigh flaps (ALT) requires precise perforator selection and accurate tissue perfusion assessment. Unfortunately, variable perforator anatomy and perfusion patterns often result in extensive exploratory dissection. We aimed to assess the impact of preoperative color-coded duplex sonography (CCDS) and intraoperative indocyanine green fluorescence angiography (ICGFA) on perforator selection and operative morbidity. Fifty-three ALTs were performed with preoperative CCDS and intraoperative ICGFA. Flaps had one, two, or three suitable perforators. Additional perforators were either included, or ligated following temporary clamping with ICGFA-based perfusion assessment. If perfusion was sufficient, further dissecting of additional perforators of unfavorable course was abstained from. The impact on perforator selection and operative outcomes was studied. Seven flaps were raised on a single, 34 on 2, and 12 on 3 perforators. There was no flap loss. Comparing the subgroups of fully dissected versus partially clamped and subsequently ligated perforators revealed significantly shorter harvest times in the latter (268 ± 71 versus 216 ± 47 min, *p* = 0.006). The unnecessary dissection of 21 additional perforators in 16 cases was avoided. Combining preoperative CCDS and intraoperative ICGFA aids in designing ALTs and guarantees the intraoperative selection of suitable perforators. This allows for significant reductions in operative time and donor-site morbidity by limiting unnecessary dissection.

## 1. Introduction

The anterolateral thigh (ALT) perforator flap is a workhorse flap in microsurgical reconstruction [1]. It is raised on septocutaneous, myocutaneous, or septomyocutaneous perforating branches of the descending branch of the lateral circumflex femoral artery [2]. However, the high perforator variability can pose significant challenges, as tedious transmuscular vessel dissection prolongs operative time and increases donor-site morbidity due to considerable neuromuscular disintegration [3,4,5,6]. In addition, due to intra- and interindividual variations in skin perfusion patterns, it is often difficult to discern individual perforasomes intraoperatively [7,8]. Thus, intraoperative perforator selection can be impeded, as the microsurgeon faces the question of which and how many perforators should be included when raising the flap. Inexperienced microsurgeons in particular may tend to include as many perforators as possible to ensure maximum tissue perfusion, thus leading to prolonged operative times.

We recently showed that preoperative color-coded duplex ultrasonography (CCDS)-assisted perforator mapping was associated with significantly reduced ALT free flap harvesting and operative times [9]. Furthermore, the respective microsurgeons relied on single-perforator flaps significantly more often in preoperatively duplexed patients [10]. As another perioperative technical innovation, indocyanine green fluorescence angiography (ICGFA) has been adopted in free flap surgery for imaging of tissue perfusion by capturing the fluorescence of the intravenously administered contrast agent indocyanine green (ICG) [11,12,13,14]. ICGFA is particularly helpful in precisely designing perforator flaps by balancing intraoperative tissue perfusion and flap geometry [15,16]. Numerous clinical studies have since commented on the beneficial effects of intraoperative ICGFA and have advocated for its potential for improving clinical outcomes in reconstructive microsurgery [11,12,13,14,15,16,17,18].

Given the challenges of variable perforator anatomy and inspired by technological advancements, we refined our approach by combining both preoperative perforator mapping via CCDS and intraoperative perforator selection via ICGFA. This study aimed to analyze our refined approach to ALT free flap harvesting. The primary goal was to assess the impact of ICGFA on the decision-making process regarding intraoperative ALT perforator selection as well as to highlight the associated intra- and postoperative benefits. We theorized that by combining preoperative CCDS and intraoperative ICGFA, flap design and perforator dissection could be refined, shifting from solely anatomical to functionally-driven decisions. Specifically, we hypothesized that this integrated approach could enhance perforator selection efficiency by direct in vivo visualization of the corresponding tissue perfusion, enabling a sufficiency-based strategy rather than maximalist dissection.

## 2. Materials and Methods

### 2.1. Patients

This retrospective cohort study included all consecutive patients scheduled for and undergoing soft tissue reconstruction using ALT free flaps at our institution between January 2017 and June 2019. Inclusion criteria were as follows: adult patients requiring ALT free flap reconstruction for soft tissue defects and provision of written informed consent for the procedure. Exclusion criteria were as follows: refusal of written informed consent for the procedure and known allergies or intolerances to indocyanine green (ICG).

### 2.2. Study Design and Setting

Between January 2017 and June 2019, 53 consecutive ALT free flaps were performed at the BG Trauma Center in Ludwigshafen, Germany, a tertiary care and university teaching hospital, in adherence to our refined approach with preoperative CCDS-assisted perforator mapping complemented by intraoperative perfusion assessment via ICGFA. All surgeries were performed by board-certified plastic surgeons, senior microsurgeons at our institution, with over 10 years of experience and triple-digit case numbers in free flap surgery. The present study and corresponding protocol were approved by the local ethics committee of Rhineland-Palatinate (Mainz, Germany, registration number: 2022-16484). Controls and cases were retrospectively identified and included in this study; data were then obtained from retrospective chart review. This retrospective cohort study is being reported in line with the 2021 STROCSS (Strengthening the reporting of cohort studies in surgery) guidelines [19].

### 2.3. Methods

Prior to ALT free flap reconstruction, all patients underwent standardized preoperative optimization. This included thorough debridement of the recipient site, with microbiological sampling obtained to guide targeted antibiotic therapy when indicated. CT- or MR-angiography was routinely performed to assess recipient-vessel patency, and interventional or surgical revascularization was undertaken if deemed necessary by the multidisciplinary team. Routine preoperative blood work and hypercoagulability screening were also performed to identify and address any potential risk factors.

Our standardized ALT free flap harvest procedure consisted of preoperative localization of suitable perforators using CCDS [9,10]. ALT free flaps were then raised, centered around the identified perforators in the subfascial plane according to standard principles with a maximum of three perforators categorized into the following groups: group 1 (one perforator), group 2 (two perforators), and group 3 (three perforators). Intraoperatively, according to the surgeon’s judgment, the most suitable perforator was chosen and dissected in retrograde fashion to its origin from the lateral circumflex femoral artery. Subsequently, flap harvest was completed either by including additional perforators based on the primary microsurgeon’s preference (subgroups 2A and 3A, controls) or by deliberately discarding them after sufficient tissue perfusion had been ascertained via ICGFA (subgroups 2B and 3B, cases). In the latter cases, all additional perforators of unfavorable transmuscular course were temporarily clamped and ICGFA was carried out before continuing the dissection. If flap perfusion was sufficient on a single perforator according to ICGFA, those temporarily clamped perforators were then definitely ligated (subgroups 2B and 3B). A schematic overview of the groups is shown in Figure 1.

The primary intraoperative outcomes were overall operative and flap harvest times, as well as the number of dissected perforators. The primary postoperative outcomes were uneventful healing versus complications due to inadequate tissue perfusion requiring reoperation under general anesthesia (corresponding to Grade IIIb on the Clavien–Dindo classification [20]). Clinical perfusion findings, ICGFA perfusion findings, and details of the decision-making were documented in all cases.

Standardized postoperative care protocols were implemented for all patients. This included a period of five days of bed rest with elevation of the recipient body part in cases of extremity reconstruction, or avoidance of pressure on the recipient site for torso defects. All patients received prophylactic subcutaneous administration of 40 mg Clexane^®^ (enoxaparin) twice daily for five days postoperatively. For lower extremity reconstructions, a standardized dangling procedure was initiated after the bed rest period, and prophylactic enoxaparin was continued until patients achieved normal ambulation. Postoperative follow-up was conducted daily during the inpatient stay. Following discharge, patients were scheduled for outpatient follow-up visits at 1 and 6 weeks postoperatively.

### 2.4. Operative Technique of ICGFA-Assisted ALT Perforator Selection

ALT flap design and markings were performed according to standard principles after the preoperative localization of perforators using CCDS as described elsewhere [9,10]. Flap harvesting was begun via incision at the medial flap down to and through the thigh muscle fascia. Dissection was carried out in the subfascial plane laterally and suitable skin perforators were exposed (Figure 2). Next, the descending branch of the lateral circumflex femoral artery (LCFA) was visualized within the intermuscular septum between the rectus femoris and the vastus lateralis muscles. If two or more perforators were present, the best-suited perforator according to its size, pulsatility, position, and course was chosen and dissected to the descending branch of the LCFA (Figure 3). Septocutaneous perforators were favored over intramuscular perforators in order to avoid transmuscular dissection through the vastus lateralis muscle and minimize donor-site morbidity. In subgroups 2B and 3B, additional perforators of unfavorable morphology or transmuscular course were then temporarily clamped (Figure 4), and the flap incision completed (Figure 5). After ICGFA-assisted confirmation of sufficient flap perfusion through the single unclamped and isolated perforator (Figure 6), all previously clamped perforators were ligated.

### 2.5. Technical Details of ICGFA-Assisted Perfusion Assessment

Intraoperative ICGFA was performed using the Fluobeam^®^ 800 system (Fluosoft® version 2.3 software, Fluoptics, Grenoble, France) according to the following standardized protocol in order to guarantee interventional consistency: After the complete circumcision of the fasciocutaneous flaps, tissue perfusion was first assessed clinically (capillary blanching and refill, skin color and turgor, as well as dermal bleeding) in a standardized manner after a five-minute run-in period of the flap with the mean blood pressure raised above 70 mmHg through the administration of cafedrine/theodrenaline (Akrinor^®^, Ratiopharm, Ulm, Germany). Further vasoactive medications were not administered at the time of perfusion assessment. Indocyanine Green (Verdye^®^, Diagnostic Green GmbH, Aschheim-Dornach, Germany) was then dissolved in sterile water and administered intravenously as a bolus via a central or peripheral venous catheter at 0.15 mg/kg body weight per injection. The ICG fluorescence intensity was recorded and displayed in real-time as a grayscale image directly after intravenous administration. Images were immediately assessed with the Fluosoft^®^ version 2.3 software (Fluoptics, Grenoble, France).

### 2.6. Statistical Analysis

Formal sample size calculation was not performed prior to this retrospective cohort study due to the lack of pre-existing data on the combined use of preoperative CCDS and intraoperative ICGFA for ALT free flaps. Instead, we included a consecutive series of patients undergoing ALT free flap reconstruction during the specified study period. Our sample size is therefore determined by the number of patients treated with this approach during this timeframe. The decision to implement this combined approach as our standard of care was subsequently informed by the observed benefits in this consecutive series, which justified further retrospective analysis. Continuous interval-scaled variables are reported as arithmetic means with standard deviations and ranges. Ordinal and dichotomous data are reported as quantities with percentages. All statistical analyses were performed with Graphpad Prism Version 8 (Graphpad Software, Inc., La Jolla, CA, USA). The unpaired Welch-corrected t-test was employed to assess differences in means for continuous variables of interest. Subgroup analyses were planned a priori to compare outcomes between subgroups 2B and 3B (ICGFA-guided selection) and subgroups 2A and 3A (surgeon’s preference). An error probability of *p* < 0.05 was considered statistically significant.

## 3. Results

The perforators of 53 ALT free flaps were identified preoperatively via CCDS and the respective flaps were designed using defect-specific templates centered around them. The respective tissue perfusion was then assessed intraoperatively via ICGFA in adherence to the afore-stated study protocol. No intolerances or allergic reactions to ICG were observed. The mean size of raised ALT flaps was 21.8 × 8.3 cm (range: 6.0 × 4.0 to 32.0 × 15.0 cm). Regarding the number of available perforators amongst the entirety of all 53 ALT flaps, there was a single perforator available for dissection in seven cases (group 1, 13.2%), two available perforators in 34 cases (group 2, 64.2%), and three available perforators in 12 cases (group 3, 22.6%). Figure 1 illustrates a schematic overview of all three groups. Causes for reconstruction were traumatic injuries in 38 patients (71.7%), wound infections and chronic wounds in 11 patients (20.8%), and oncologic defects in four patients (7.5%). The lower extremity was the most common recipient site (*n* = 38; 71.7%), followed by the upper extremity (*n* = 9; 17.0%), and the torso (*n* = 6; 11.3%). Patient characteristics and clinical data are summarized in Table 1 and Table 2. The postoperative course was uneventful in 49 patients (92.5%). While there was no total flap loss, four ALT flaps (7.5%) developed partial necroses of the proximal (*n* = 2) and distal tips (*n* = 2) during the immediate postoperative course. There were no significant differences in group incidences. Among these patients, three required surgical redebridement and either flap readvancement (*n* = 1) or skin grafting (*n* = 2) for secondary closure. Notably, in all cases of partial necrosis, the flap design was overextended beyond the indicated perforasome perfusion area as indicated by ICGFA assessment.

### 3.1. Single-Perforator ALT Flaps (Group 1)

A total of seven ALT free flaps (13.2%) had only one available perforator (group 1) and were raised accordingly. The mean ALT flap size was 22.0 × 9.1 cm (range: 17.0 × 6.0–32.0 × 15.0 cm). All flaps healed uneventfully, and no postoperative complications occurred. The overall operative time was 423 ± 111 min, and the overall flap harvest time was 233 ± 54 min. Clinical data are summarized in Table 1.

### 3.2. Two-Perforator ALT Flaps (Group 2)

Out of 34 ALT free flaps (64.2%) with two available perforators (group 2), 23 flaps were raised including both perforators (group 2A). The mean ALT flap size was 22.0 × 7.9 cm (range: 17.6 × 4.0–28.0 × 12.0 cm). The corresponding overall operative time was 446 ± 115 min with a corresponding overall flap harvest time of 269 ± 75 min. A total of 11 ALT flaps, however, were raised on only one of the two available perforators in adherence to the afore-stated ICGFA-based perforator selection (group 2B). The corresponding overall operative time was reduced to 420 ± 103 min with the corresponding overall flap harvest time being reduced to 212 ± 54 min (*p* = 0.02). The patient characteristics and operative details per subgroup are summarized in Table 2 and Table 3.

### 3.3. Three-Perforator ALT Flaps (Group 3)

A total of 12 ALT free flaps (22.6%) had three available perforators (group 3). Of these, seven flaps were raised including all three available perforators as per the primary microsurgeon’s preference (subgroup 3A). The mean ALT flap size was 21.8 × 9.0 cm (range: 16.0 × 5.0–28.0 × 12.0 cm). The overall operative time was 428 ± 82 min and the overall flap harvest time was 265 ± 60 min. A subgroup of five ALT flaps, however, were raised on only one perforator in adherence to the ICGFA-based algorithm (subgroup 3B). The corresponding overall operative time was 412 ± 25 min, and the overall flap harvest time was 226 ± 27 min. The patient characteristics and operative details per subgroup are summarized in Table 2 and Table 4.

### 3.4. Comparison of Flap Harvest Times Between ICGFA-Assisted Perforator Selection Versus Conventionally Raised ALT Flaps

The overall flap harvest times were 233 ± 54 min in group 1, 228 ± 60 min in group 2, and 249 ± 50 min in group 3. Comparing the entirety of ALT free flaps with fully dissected perforators (subgroups 2A and 3A) versus those with ICGFA-assisted selection of a single dominant perforator (2B and 3B), revealed significantly shorter harvest times in the latter subgroups (268 ± 71 min in subgroups 2A and 3A versus 216 ± 47 min in subgroups 2B and 3B, mean difference 52 ± 18 min, *p* = 0.006; Figure 7, Table 5). In detail, the mean flap harvest time was 58 ± 23 min shorter in subgroup 2B (212 ± 54 min) versus 2A (269 ± 75 min, *p* = 0.02) and 39 ± 29 min shorter in subgroup 3B (226 ± 27 min) versus 3A (265 ± 60 min, *p* = 0.15). In sum, the mean flap harvest time was significantly lower when the ICGFA-based algorithm for perforator selection was adhered to (subgroups 2B and 3B, *p* = 0.006).

Following the ICGFA-based perforator selection algorithm enabled refraining from pursuing 21 additional perforators. Thus, this approach helped avoid unnecessary vessel dissection in 16 thighs, resulting in reduced flap harvest time as well as donor-site morbidity. Complications or adverse events related to ICG administration were not observed.

## 4. Discussion

In microsurgical reconstruction, the primary emphasis has been transitioning from simple defect coverage to optimizing various perioperative aspects [10,21,22]. Amongst these, the reduction of operative time has been shown to effectively decrease perioperative complication rates and overall treatment costs, making it a matter of both substantial medical as well as economic interest [23,24]. To align with this shifting focus, we have introduced the afore-stated intraoperative ALT perforator selection protocol in recent years. By comparing 16 cases of intraoperative ICGFA-based perforator selection versus 30 conventionally raised controls in the present retrospective single-center cohort study, we were able to demonstrate a statistically significant reduction of flap harvest times when this algorithm was followed.

In the context of vessel selection, the use of ICGFA to assess the distal perfusion of a lower extremity prior to harvesting a thigh perforator flap based on either the transverse or descending branch of the lateral circumflex femoral artery has been described by Maxwell in order to rule out distal limb ischemia in a patient with severe peripheral artery disease [25]. La Padula and colleagues were the first to apply intraoperative ICGFA during ALT flap harvest to aid in the decision to ligate additional perforators in a retrospective study of 28 patients with lower extremity defects [26]. Moreover, ICGFA has recently also been used to evaluate recipient vessels in lower extremity reconstruction [27].

Nevertheless, since its adoption in free flap surgery, ICGFA has primarily been utilized to image overall tissue perfusion, aiming to reduce postoperative complication rates by visualizing the areal extent of flap perfusion [16]. Over the course of our previous studies focusing on ICGFA-based ALT free flap perfusion assessment, we further expanded its application to incorporate the perforator selection process. While ICGFA has thus far been primarily described as a method to assess flap perfusion after the completion of harvest—i.e., identifying insufficiently perfused areas warranting prophylactic trimming prior to transfer—we implemented its additional use at an earlier stage during dissection. During flap raise, it can be difficult to discern which and how many perforators should be included, with considerable implications for the remaining flap harvest time and overall strain on the respective donor thigh. The aim of this study was therefore to analyze our refined approach to ALT free flap harvesting and to assess the impact of ICGFA on intraoperative perforator selection.

To this end, ICGFA helped minimize donor-site morbidity by facilitating the intraoperative decision in favor of a single yet sufficient perforator, thus allowing the respective microsurgeon to refrain from the unnecessary dissection of additional perforators. Not only did this approach save valuable operative time, thereby increasing overall efficiency, but it also restricted surgical invasiveness, thus holding the potential to increase patient safety. We therefore believe that La Padula’s approach, involving “dissection of both vessels […] to provide a better flap perfusion […] if two adjacent dominant perforators [are] identified” and subsequent ICGFA [26], should be further refined to incorporate ICGFA-based perforator selection as early as possible before any additional perforators are dissected free at all.

While our observations offer new insights into the advantages of intraoperative ICGFA-based ALT perforator selection, it is essential to critically assess our study in consideration of its methodical shortcomings. First, our study faces important limitations due to its retrospective single-center design, which introduces the potential for biased case and control selection. Furthermore, the study cohort comprised only a relatively small sample size with no random patient allocation to either group. As a result, our findings allow only for the suggestion of associations, but do not establish real causal relationships. Therefore, the generalizability of our findings has to be questioned. In addition, the small sample size jeopardized the subgroup analysis of the three-perforator flap cohort (subgroups 3A and 3B), which failed to reach statistical significance. Larger cohort studies and prospective trials, as well as investigations of other perforator flaps are therefore needed. Third, it is important to note that all ALT harvest times were measured from the initial skin incision to the onset of flap ischemia, as recorded in the operative reports. Other potential surgical procedures preceding flap elevation, including wound debridements, fracture fixations, or prolonged recipient-vessel dissections, could have influenced the duration of this parameter. Moreover, the importance of an individual training level and comfort with vessel selection and subsequent dissection cannot be overstated in a microsurgeon’s decision to pursue or refrain from pursuing a specific perforator. This is particularly evident in an academic training center like ours—an aspect that remains uncontrolled and cannot be successfully blinded. Finally, whether the unnecessary dissection of further perforators does indeed increase donor-site morbidity and whether refraining from pursuing additional dissections reduces it in turn, remains speculative—albeit likely.

## 5. Conclusions

The integration of preoperative CCDS with intraoperative ICGFA not only streamlines the design of single-perforator ALT free flaps but also ensures the efficient selection of dominant perforators during surgery. This functional guidance optimizes surgical resource allocation by reducing operative time through minimized unnecessary dissection, as well as donor-site morbidity due to less neuromuscular disruption. Thus, the combined use of CCDS and ICGFA optimizes the harvesting of free ALT flaps and contributes to a more favorable outcome with increased efficiency and reduced donor-site morbidity. Future research should validate these tenets and explore broader applications for other perforator flaps to lay the foundation for advancing reconstructive microsurgery towards more efficient and less invasive techniques.

## Figures and Tables

**Figure 1 life-15-00620-f001:**
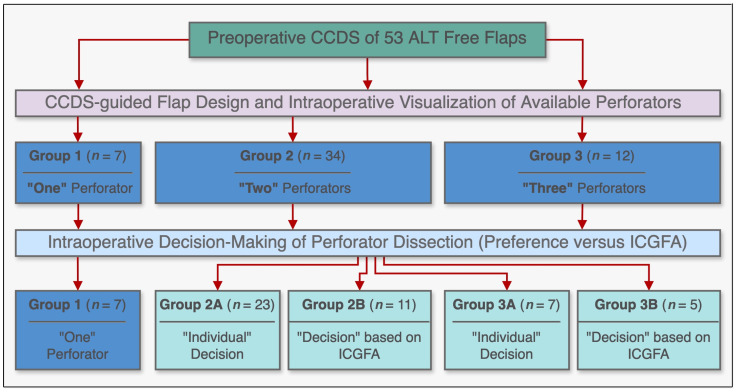
Schematic diagram illustrating the study subgroups and their respective methodologies, incorporating preoperative color-coded duplex ultrasonography (CCDS) and intraoperative indocyanine green fluorescence angiography (ICGFA) for ALT free flap design and harvest.

**Figure 2 life-15-00620-f002:**
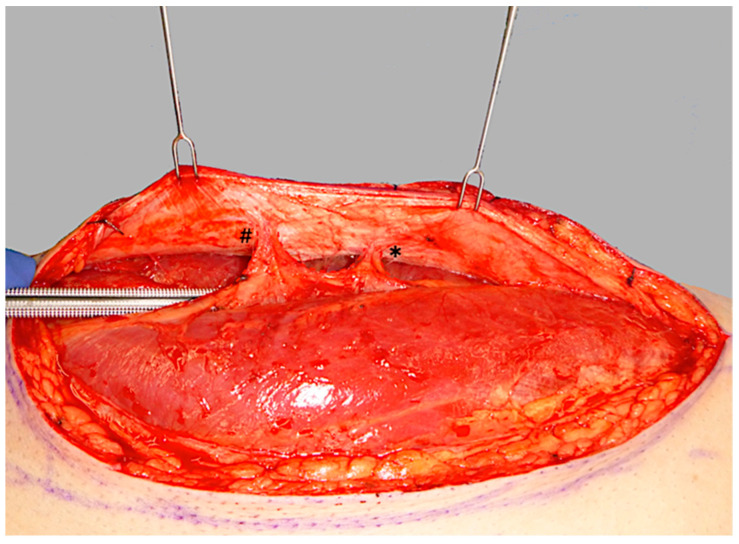
Initial subfascial dissection from the medial flap border, exposing both a proximal perforator (*) and a distal perforator (#).

**Figure 3 life-15-00620-f003:**
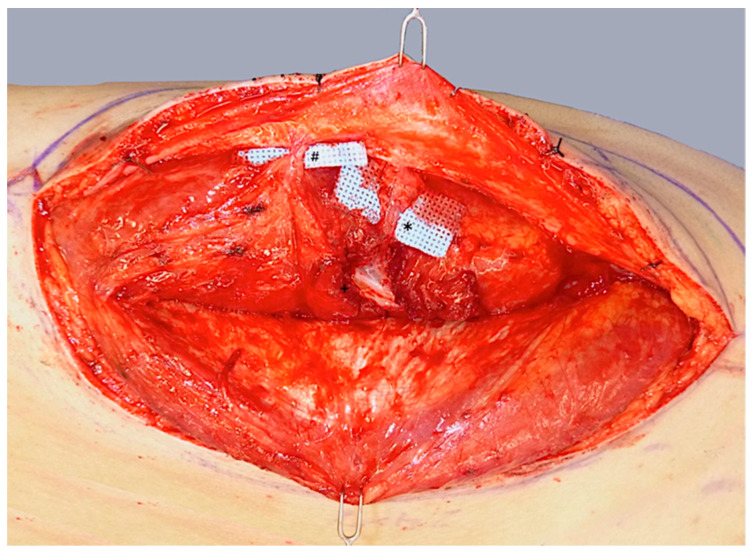
Dissection of a central dominant septomyocutaneous perforator (*) with a short transmuscular course through the vastus lateralis muscle to its origin at the descending branch of the lateral circumflex femoral artery (LCFA; +). A less prominent distal perforator (#) with a longer intramuscular course is visualized but not dissected.

**Figure 4 life-15-00620-f004:**
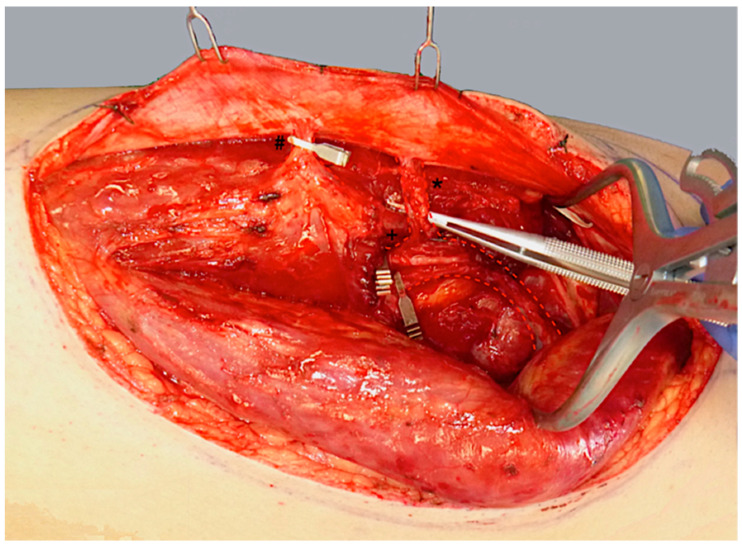
Temporary clamping of the distal perforator (#) and the descending branch of the LCFA (+) distal to the proximal perforator (*). This isolates flap perfusion to the central, proximal perforator (*) to assess its sufficiency.

**Figure 5 life-15-00620-f005:**
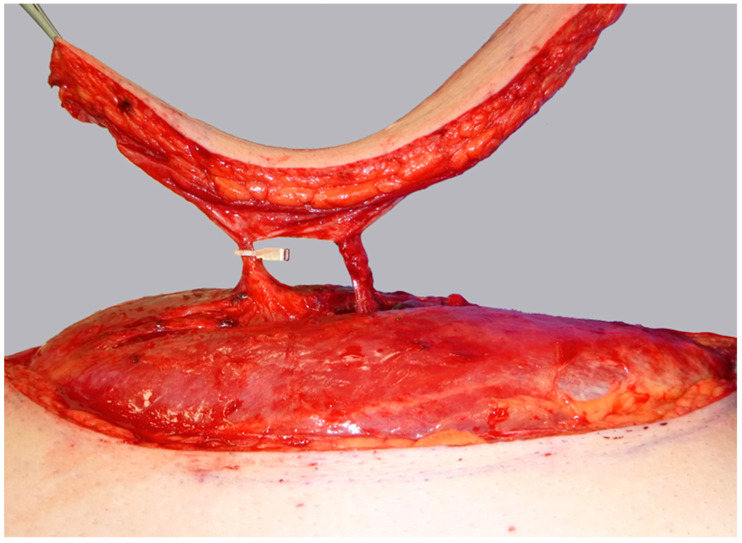
Flap harvest is completed while the distal perforator remains temporarily clamped, awaiting ICGFA assessment.

**Figure 6 life-15-00620-f006:**
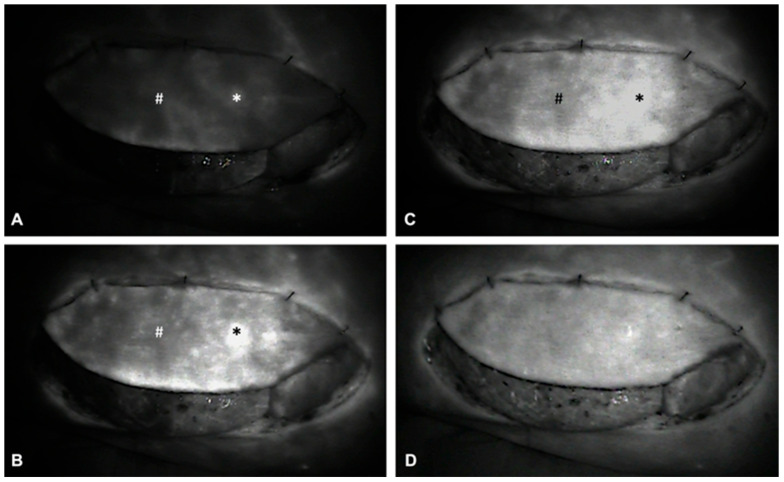
Intraoperative ICGFA demonstrating sequential flap perfusion via the proximal perforator (*) prior to distal perforator (#) ligation. Panels (**A**–**D**) illustrate the following: (**A**) Early arterial perfusion of the flap is visualized via the proximal perforator (*). (**B**) ICG perfusion extends to the periphery surrounding the proximal perforator (*). (**C**) Progressive perfusion continues, visualizing the perfusion of the remaining flap tissue. (**D**) Homogenous ICG uptake and perfusion of the entire flap confirms adequate flap perfusion via the proximal perforator (*), allowing for subsequent ligation of the temporarily clamped distal perforator (#).

**Figure 7 life-15-00620-f007:**
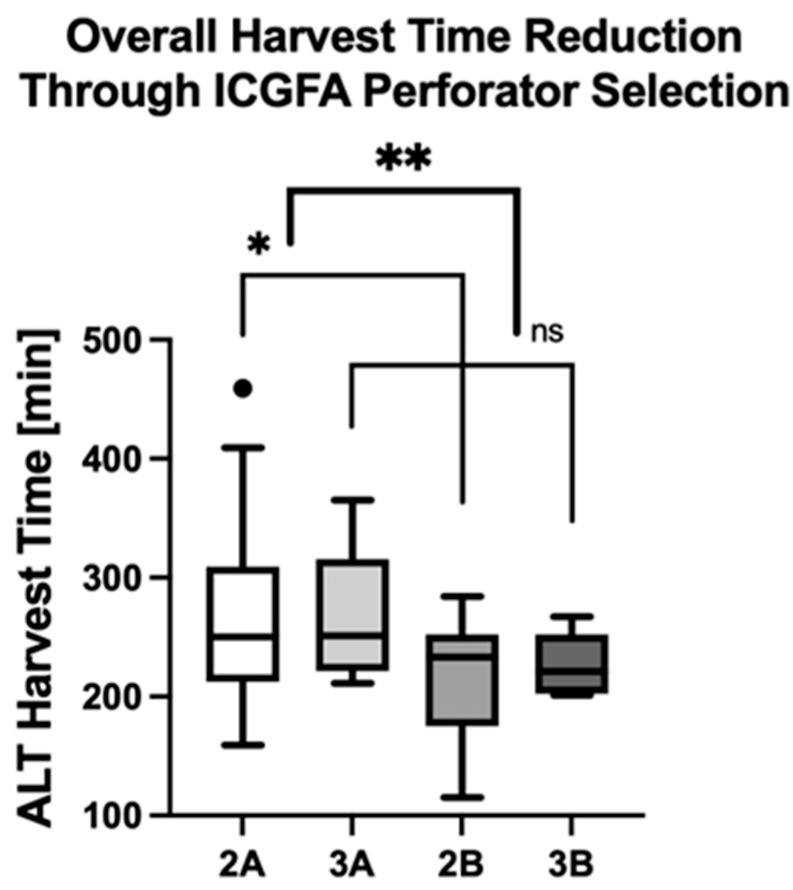
ICGFA-assisted perforator selection significantly reduces ALT flap harvest times. Asterisks indicate statistically significant differences (* *p* = 0.02, ** *p* = 0.006).

**Table 1 life-15-00620-t001:** Patient demographics and defect characteristics of the study cohort undergoing ALT flap reconstruction.

Total number of ALT free flaps	53
Gender (female/male)	9/44
Mean age [years] ± SD (range)	48.7 ± 16.9 (18.0–83.0)
Mean flap size [cm] (range)	21.8 × 8.3 (6.0 × 4.0–32.0 × 15.0)
*Cause for reconstruction*	
Traumatic defect [n] (%)	38 (71.7%)
Infectious/chronic wound [n] (%)	11 (20.8%)
Oncologic defect [n] (%)	4 (7.5%)
*Flap recipient site*	
Lower extremity [n] (%)	38 (71.7%)
Upper extremity [n] (%)	9 (17.0%)
Torso [n] (%)	6 (11.3%)

**Table 2 life-15-00620-t002:** Patient characteristics and operative details categorized by study group (group 1, group 2, and group 3).

	Group 1 (1 Perforator)	Group 2 (2 Perforators)	Group 3 (3 Perforators)
Number of patients [n] (%)	7 (13.2%)	34 (64.2%)	12 (22.6%)
Gender (male/female)	6/1	27/7	12/2
Mean age [years] ± SD (range)	40.7 ± 20.9 (18–73)	47.6 ± 19.6 (18–83)	48.9 ± 16.7 (27–73)
Mean flap size [cm], range	22.0 × 9.1, 17.0 × 6.0–32.0 × 15.0	22.0 × 7.9, 17.6 × 4.0–28.0 × 12.0	21.8 × 9.0, 16.0 × 5.0–28.0 × 12.0
Mean operative time ± SD [min] (range)	423.3 ± 110.8 (255–539)	437.6 ± 110.3 (275–823)	414.6 ± 77.6 (287–551)
Mean flap harvest time ± SD [min](range)	233.0 ± 53.9 (145–239)	249.4 ± 72.8 (115–459)	245.5 ± 48.2 (201–365)

**Table 3 life-15-00620-t003:** Subgroup analysis of group 2 (two available perforators): Comparison of subgroups 2A (both perforators dissected) and 2B (ICGFA-guided single-perforator selection).

	Subgroup 2A	Subgroup 2B	*p*-Value
Number of dissected perforators	2	1	
Number of patients (%)	23 (67.6%)	11 (32.4%)	
Mean flap size [cm]	22.0 × 9.1	22.0 × 7.9	
Mean operative time ± SD [min] (range)	445.8 ± 114.9 (279–823)	420.4 ± 103.2 (275–597)	>0.05
Mean flap harvest time ± SD [min] (range)	269.2 ± 74.6 (159–459)	211.5 ± 53.9 (115–284)	0.02

**Table 4 life-15-00620-t004:** Subgroup analysis of group 3 (three available perforators): Comparison of subgroups 3A (all three perforators dissected) and 3B (ICGFA-guided single-perforator selection).

	Subgroup 3A	Subgroup 3B	*p*-Value
Number of dissected perforators	3	1	
Number of patients (%)	7 (58.3%)	5 (41.7%)	
Mean flap size [cm]	24.0 × 9.8	18.6 × 8.0	
Mean operative time ± SD [min] (range)	428.3 ± 81.8 (295–551)	412.0 ± 24.7 (376–442)	>0.05
Mean flap harvest time ± SD [min] (range)	265.0 ± 59.6 (211–365)	226.0 ± 27.1 (201–267)	>0.05

**Table 5 life-15-00620-t005:** Comparative analysis of flap harvest times: Fully dissected perforator subgroups (2A and 3A) versus ICGFA-assisted single-perforator selection subgroups (2B and 3B).

	Subgroups 2A and 3A	Subgroups 2B and 3B	*p*-Value
Number of patients (%)	23 + 7 (65.2%)	11 + 5 (34.8%)	
Mean flap harvest time ± SD [min] (range)	268.4 ± 70.9 (159–459)	216.1 ± 46.7 (115–284)	0.006

## Data Availability

Data are available from the first and corresponding authors upon reasonable request.

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
