# Peer review of "Optimizing ALT Flap Harvest: The Role of Combined Preoperative Duplex Ultrasound and Intraoperative ICG Angiography for Perforator Selection"

_life, 2025, doi:10.3390/life15040620_

Round 1

Reviewer 1 Report

Comments and Suggestions for Authors

The manuscript “Combining preoperative color-coded duplex ultrasonography  and intraoperative indocyanine green fluorescence angiography  for optimizing anterolateral thigh perforator selection and flap  harvest – a single-center retrospective cohort study” by Thomas et al. is devoted to the optimization of free flap harvesting. It can be of interest to a broad clinical audience.

The study is well-planned and well-described. It can be published if the following minor deficiencies are addressed:

  1. The discussion needs to be streamlined. Many parts of the discussion (e.g., Lines 301-307) belong to the Introduction and repeat what was said already.
  2. Lines 334-336: I don’t understand what the authors meant by “dissection of both vessels […] to provide a better flap perfusion […] if two adjacent dominant perforators [are] identified”. Is that a citation? It looks odd.
Comments on the Quality of English Language

no major concerns

Author Response

Thank you for your time and your feedback on our manuscript. We have made one revision in response to remark 1 and would like to offer a clarifying comment on point 2, as detailed below:

Comments 1: The discussion needs to be streamlined. Many parts of the discussion (e.g., Lines 301-307) belong to the Introduction and repeat what was said already.

Response 1: The indicated repetitive section has been removed from the discussion, as suggested.

Comments 2: Lines 334-336: I don’t understand what the authors meant by “dissection of both vessels […] to provide a better flap perfusion […] if two adjacent dominant perforators [are] identified”. Is that a citation? It looks odd.

Response 2: This is not a citation, but indeed a direct quote. And the way it looks is in adherence with scientific standards as well as the MDPI guidelines (see the MDPI Layout Style Guide, for example)

Reviewer 2 Report

Comments and Suggestions for Authors

SUMMARY

This study presents a retrospective analysis evaluating the impact of combining preoperative color-coded duplex ultrasonography (CCDS) with intraoperative indocyanine green fluorescence angiography (ICGFA) on anterolateral thigh (ALT) free flap harvesting. The authors hypothesize that this combined approach help flap planning and perforator dissection, in order to reduced operative time and donor-site morbidity

TITLE

Reformulate the title in a shirter way: e.g., "Precision Flap Harvest: Optimizing ALT Perforator Selection with Duplex Ultrasound and ICG Angiography"

ABSTRACT

well-written and effectively summarizes the study

"34 on two, and 12 on three" USE LETTERS OR NUMBERS

MAIN STRENGHTS:

The study focuses on a clinically relevant problem (optimizing ALT flap harvesting to reduce operative time and donor-site morbidity)

The introduction describes the challenges related to ALT flap surgery, especially the variability of perforator anatomy in an effective manner

The methods section includes a detailed description of the study design, patient selection, and surgical technique.

The use of both preoperative and intraoperative imaging techniques is well explained

The results are well- presented with numerical data, including operative times and flap harvest times.

The discussion effectively describes the results ffrom the existing literature. The clinical implications of the findings are provided.

LIMITATIONS

The sample size, particularly in the three-perforator group, is small. This limits the statistical power of the subgroup analysis.

The selection of the "best-suited" perforator is based on the surgeon's opninio, resulting subjectivity

Author Response

We greatly appreciate you taking the time to review our manuscript. In accordance with your suggestions, we have implemented the following revisions:

Comments 1: Reformulate the title in a shirter way: e.g., "Precision Flap Harvest: Optimizing ALT Perforator Selection with Duplex Ultrasound and ICG Angiography"

Response 1: The title was shortened to “Optimizing ALT flap harvest: The role of combined preoperative duplex ultrasound and intraoperative ICG angiography for perforator selection”

Comments 2: "34 on two, and 12 on three" USE LETTERS OR NUMBERS

Response 2: As suggested, the abstract was changed to include only numbers.